# Modified Natural Dolomite and Its Influence on the Production of Glycerol Carbonate: Effects of Structural and Basicity Properties

**DOI:** 10.3390/ma14092358

**Published:** 2021-05-01

**Authors:** Julio González-García, Lifang Chen, Omar Campuzano-Calderon, Sara Núñez-Correa, Enrique A. López-Guajardo, Jin An Wang, Alejandro Montesinos-Castellanos

**Affiliations:** 1Tecnologico de Monterrey, Escuela de Ingeniería y Ciencias, Ave. Eugenio Garza Sada 2501, Monterrey, Nuevo León 64849, Mexico; a00824173@itesm.mx (O.C.-C.); sarnunez@uv.mx (S.N.-C); enrique.alopezg@tec.mx (E.A.L.-G.); 2Tecnologico de Estudios Superiores de Coacalco, 16 de septiembre 54, Col. Cabecera municipal, Coacalco de Berriozabal, Estado de México 55700, Mexico; 3ESIQIE, Instituto Politécnico Nacional, Av. Instituto Politécnico Nacional s/n, Col. Zacatenco, Ciudad de México 07738, Mexico; jwang@ipn.mx; 4Facultad de Ciencias Químicas, Universidad Veracruzana, Campus Coatzacoalcos, Veracruz 96538, Mexico

**Keywords:** crystalline composition, natural dolomite, glycerol carbonate, acid treatments, surface basicity

## Abstract

A systematic study over different treatment conditions, including hydrothermal and acid-thermal, was successfully carried out to determine the most suitable conditions to enhance the textural properties and surface chemical composition of natural dolomite. The reconstruction of dolomite after various treatments enhanced the surface area by 4–5 times and diminished the pore diameter between 70% and 81% compared to the untreated parent dolomite. The Rietveld analysis of the X-ray diffraction (XRD) patterns revealed changes in the crystalline compositions after each treatment. When the treated dolomite was used as a catalyst to produce glycerol carbonate via a transesterification reaction of glycerol and dimethyl carbonate, the crystalline Ca(OH)_2_ concentration of the modified dolomite and the apparent glycerol carbonate formation rate (r_gc_) are well-correlated. The results suggest that an increase of the crystalline Ca(OH)_2_ concentration could be related with surface basicity at the weak and moderate strength sites that may lead to an increase in catalytic activity. The hydrothermal treated dolomite showed a selectivity of glycerol carbonate greater than 99% and r_gc_ value 3.42 mmol/min·g_cat_, which was higher than that achieved on other samples. This study could aid to the proper selection of dolomite treatment for the desired crystalline composition, depending on the applications of this highly available mineral.

## 1. Introduction

Dolomite is an abundant mineral that primarily comprises double calcium and magnesium carbonates, CaMg(CO_3_)_2_. This source of calcium and magnesium is widely used as a raw material in cement and steel industries and as catalyst in biomass gasification due to its great performance, abundance and low price (~0.07 USD/kg) [1,2]. When thermally treated at temperatures greater than 800 °C, dolomite decomposes into CaO, MgO, and CO_2_ [3]. Additionally, the calcined dolomite has been used as a CO_2_ sorbent in looping cycles due to its high calcium content. Even though its CO_2_ sorption capacity (0.12–0.50 g_CO2_/g_sorbent_) is comparable with those of other calcium-based sorbents, its surface area is lower than 20 m^2^/g and its CO_2_ capture capacity decreases as the number of calcination–carbonation cycles increases. This is mainly caused by a sintering effect and a decrease in surface area [4]. In order to overcome these drawbacks and to increase the adsorption performance of natural sorbents, such as dolomite and limestone, various acid treatments have been proposed.

For instance, Hu et al. [5] modified the calcined limestone (CaCO_3_) with excess quantities of different organic acids (i.e., tartaric, acetic, propionic, and citric acids). Sun et al. [6] treated the calcined limestone with excess propionic acid (100% more of the stoichiometric molar ratio with respect to CaO) to improve the ability of capturing CO_2_. Their results indicated an increase in the surface area by 23.1% and 150.2%, respectively, while the number of CO_2_ capture cycles improved. Both authors observed an increase in micropores generation after acid treatment and calcination of limestone. Hu et al. [5] reported an average pore size reduction from 50 to 4 nm. The enhanced textural properties and thus, the increase of CO_2_ capture capacity could be attributed to both, the rearrangement of the crystalline structure and the ability of resistance to sintering of solids propionic acid treatment. Moreover, Li et al. [7] reported that treating dolomite with acetic acid at different calcination temperatures during several carbonation/calcination cycles reduced the grain size and improved the carbonation performance due to an acetification process. The authors demonstrated that acid treatments can influence in the textural and physicochemical properties of dolomite.

Although the enhancement of dolomite textural properties by acid treatment has been widely explored, most of those studies focused on the investigation at a fixed acid concentration without considering the pH value of the acid solution. The interaction of acids with the dolomite may generate dissolution by various mechanisms, leading to changes in the surface charge, solute precipitation as well as the formation of intermediaries that modify its structural properties. Likewise, it is important to consider the interaction between the unreacted CaCO_3_ (residual) after calcination and the presence of H^+^ in acid solutions. This interaction could generate HCO_3_^−^ and Ca^2+^ ions. The formation of negative charges from the unpaired oxygens may benefit the creation of new hydroxyls groups. In this sense, the measurement of pH could result in an important standard parameter since it involves several factors such as the temperature, the ionic strength, the dielectric constant, the ion charge, the size of the ion, and the density of the solvent [8]. Therefore, a systematic study for evaluating the effects of acid type at different pH values on the textural and structural properties of dolomite is still required.

Similarly, the characterization of the different chemical species obtained after each acid and thermal conditions, in combination with a proper structural analysis of the treated dolomite needs to be explored. This is highly relevant to the proper and new applications in which treated dolomite could be a feasible and attractive substitute of complex sorbents and catalysts.

Given the high calcium content of dolomite, improvements on its textural properties and surface functional species by thermal and acid treatments, could make it an attractive catalyst for various reactions such as the production of glycerol carbonate. This chemical is widely used in the production of adhesives, paints, surfactants, lubricants, biodegradable polymers, lithium batteries, health care products, and membranes for gas elimination. Different reaction routes, including carbonation with phosgene, reaction with urea or CO_2_, and transesterification with organic carbonates, have been explored to produce glycerol carbonate [9]. Among the transesterification methods, the reaction of glycerol and dimethyl carbonate is very attractive since it uses non-toxic raw materials and can proceed under moderate conditions (80 °C and 1.013 bar) [10].

The apparent reaction rate of the transesterification of glycerol with dimethyl carbonate using heterogeneous catalysts, e.g., alkali or alkaline metal oxides, can be enhanced by adsorbing the reactants onto the catalyst followed by reaction between glycerol and dimethyl carbonate at the basic sites of the catalyst [11]. Although CaO is a promising heterogeneous catalyst because of its low toxicity, high activity, low cost, and availability from natural materials (e.g., limestone, dolomite, and shells), a decrease in the specific surface area of CaO has been reported during transesterification reaction. This resulted in the agglomeration and blockage of active sites and thus a significant decrease in the catalytic activity of CaO [12].

These drawbacks might be addressed by improving the chemical surface and textural properties of CaO via acid and thermal treatment and by using treated dolomite as catalyst for the transesterification reaction. Nonetheless, a thorough study is needed to determine the conditions at which the catalytic activity of dolomite could be enhanced by identifying the chemical species that contributes to the formation of basic sites needed to increase the apparent glycerol carbonate formation rate (r_gc_).

In this study, natural dolomite was treated under a variety of conditions. The crystalline phase composition, crystallite size, surface functional groups, textural properties, and surface basicity of modified dolomite were obtained after thermal and hydrothermal treatments. The influences of various acid treatments using different organic and inorganic acids at different pH values were also studied. The determination of these properties was aimed to select a proper dolomite-based catalyst to produce glycerol carbonate via the transesterification reaction of glycerol and dimethyl carbonate.

## 2. Materials and Methods

### 2.1. Dolomite and Various Treatments

Dolomite containing 43.99 wt% Ca, 11.61 wt% Mg, 0.03 wt% Na, 0.07 wt% K, and 0.07 wt% Fe balanced with C and O was provided by Grupo Calidra S.A de C.V., Mexico. The Ca/Mg ratio indicate calcitic dolomite [13]. The untreated parent dolomite (termed as U-D sample) was ground in a ball mill, sieved through a 50-mesh sieve (particle size <300 μm), and calcined at 800 °C for 4 h (termed as C-D). The calcined dolomite was treated with different organic and inorganic acid solutions purchased from Sigma-Aldrich: formic acid (95%), acetic acid (glacial), sulfuric acid (97%), nitric acid (70%), and phosphoric acid (85%). The acid solutions were prepared with deionized water for prepare four different samples, three at pH values = 1, 3, 5, and other with the proper quantity of acid to achieve a CaO + MgO stoichiometric ratio. Dolomite sample was hydrothermally treated only with deionized water at 60 °C (hydrothermal treatment, termed as C-D_H2O_).

For all acid treatments, 20 g of C-D sample was added into an Erlenmeyer flask containing 80 mL of the corresponding acid solution and stirred for 4 h at 60 °C. Afterwards, the samples were centrifuged, washed three times with deionized water, and dried overnight at 105 °C. The dried samples were again calcined at 800 °C for 6 h under static atmospheric air and the final solid was termed hydrothermal treatment sample C-D_H2O_.

The labeled dolomite samples were reported in Table 1. The first letter C or U in the sample name indicates the thermal treatment or not (C stands for calcined and U for non-calcined); the second letter D is amended with the name of dolomite; while acid used for treatment was indicated in subscript and the pH value was indicated as respective number; S indicates the stoichiometric value of (CaO + MgO) to acid molar ratio. For example, C-D_Acetic-1_ refers to the calcined dolomite treated with acetic acid at pH = 1.

### 2.2. Characterization

#### 2.2.1. XRD Analysis

The crystalline structures of all the samples were characterized by X-ray diffraction (XRD) technique in a Panalytical Empyrean X-ray diffractometer (Malvern Panalytical, Malvern, UK), with a source of Cu 40 kV and λ = 1.54 Å. The XRD patterns of the calcined samples after various treatments were recorded in a 2θ range between 10° and 70°, with a speed of 4°/min.

#### 2.2.2. Rietveld Refinement

The crystalline structures of the different dolomite samples were refined using the Rietveld method. The JAVA based software namely Materials Analysis Using Diffraction (MAUD) v2.94 was applied to refine each XRD pattern [14,15]. The estimated standard deviations or weighted profile R factors (R_wp_) are not estimated of the analysis but only of the minimum probable errors based on their normal distribution. From the Rietveld refinement, the phase compositions, crystallite size, lattice cell parameters were obtained. Each atomic fraction positions and coordinates in each crystalline structure of CaO, MgO, Ca(OH)_2_, CaCO_3_, CaSO_4_, Ca_2_P_2_O_4_, and Mg_2_P_2_O_7_ phases are reported in Supporting Information.

#### 2.2.3. Thermogravimetric Analysis (TGA)

Thermogravimetric analysis was carried out using Linseis STA PT 1600 thermal analysis instrument (Linseis, Selb, Germany) from room temperature to 1000 °C at 10 °C/min under N_2_ atmosphere, to evaluate decomposition and species elimination of the corresponding samples.

#### 2.2.4. Fourier-Transform Infrared (FT-IR) Spectroscopy

Fourier-transform infrared spectroscopy was conducted using a Perkin Elmer Frontier spectrometer (Waltham, MA, USA) in the wavelength range of 4000 to 500 cm^−1^. The resolution was 4 cm^−1^, and 64 scans were conducted for each sample. The FT-IR spectra were collected before treatment along with after treatment and drying to observe the changes of the functional groups of dolomite.

#### 2.2.5. N_2_ Adsorption-Desorption Isotherms Measurements

N_2_ adsorption-desorption isotherms experiments were performed in a Quantachrome NOVAtouch LX1 instrument (Boynton Beach, FL, USA). The samples were pre-treated by degasification at 350 °C for 12 h under vacuum. The multiple-point Brunauer–Emmett–Teller (BET) in the *P*/*P*_0_ range of 0.05–0.3 and Barrett–Joyner–Halenda (BJH) methods were used to determine the surface area and pore size distribution, respectively. Pore volume was determined as the volume of N_2_ adsorbed at *P*/*P*_0_ = 0.965.

#### 2.2.6. CO_2_ Temperature-Programed Desorption (CO_2_-TPD)

The surface basicity was measured using the temperature-programed desorption of CO_2_ (CO_2_-TPD) [16], which was carried out on a Micrometrics Autochem 2920 II instrument (Micrometrics Norcross, Norcross, GA, USA) equipped with a thermal conductivity detector. The 50-mg samples were preheated under flowing helium (30 mL/min) from room temperature to 200 °C and then kept at this temperature for 30 min. Subsequently, the samples were cooled to 70 °C, and pure CO_2_ was introduced for 30 min at 50 mL/min. The physisorbed CO_2_ was eliminated by flowing helium through the sample for 1 h. Once the thermal conductivity detector was stabilized, CO_2_-TPD was carried out from 70 °C to 900 °C at 10 °C/min under flowing helium [17].

### 2.3. Glycerol Carbonate Synthesis and Product Analysis

The glycerol carbonate synthesis via transesterification reaction was carried out with 20 g of anhydrous glycerol (99.8%, J.T. Baker), 39.44 g of dimethyl carbonate (99%, Sigma Aldrich), corresponding to a dimethyl carbonate/glycerol molar ratio of 2, and 3 wt% of catalyst based on the weight of glycerol. The catalyst and glycerol were initially added into a three-necked flask and mixed for 30 min at 600 rpm and 80 °C on a stirring hotplate. Afterwards, preheated dimethyl carbonate was added into the flask to begin the transesterification reaction. After 80 min of reaction, the solid catalyst was recovered by centrifugation at 3500 rpm and dried at 105 °C.

The products were analyzed by gas chromatography (Agilent Technologies 7820A, Agilent Technologies, Santa Clara, CA, USA; HP-5 capillary column a length of 30 m, internal diameter of 0.32 mm, and film thickness of 0.25 μm) with a flame ionization detector. Helium was used as carrier gas and fed with a flow rate of 30 mL/min. The temperature of the injector and detector was 250 °C and 270 °C, respectively. The initial column temperature of 100 °C was increased to 190 °C at a rate of 15 °C/min. The sample volume was 1 μL, and the split ratio was 50/1. Concentration was determined using the internal standard method.

## 3. Results and Discussion

### 3.1. XRD Analysis and Rietveld Refinement

The thermal, hydrothermal, and various acid treated dolomite samples were analyzed by XRD technique and their crystalline structures were refined with Rietveld method. Figure 1 shows the XRD patterns of different samples and their crystalline phase compositions, size, and lattice cell parameters obtained from the Rietveld refinement are reported in Table 2. The atomic fractional coordinates of the different crystalline structures are reported in Appendix A. Several Rietveld refinement plots are shown in Figure 2. When dolomite was thermally treated in air at 800 °C for 6 h, it contained 47.80% Ca(OH)_2_, 38.02% MgO, 13.07% CaCO_3_ and 1.10% CaO. Interestingly, the major phase in the sample was calcium hydroxide but not calcium oxide, indicating that most of the CaO was rehydrated as CaO is a very common desiccant which can adsorb water molecules from air to produce Ca(OH)_2_. It is noted that MgO showed large crystallite (3908.2 Ǻ).

After hydrothermal treatment dolomite contained 65.35% Ca(OH)_2_, 24.61% MgO, and 10.04% CaCO_3_. Calcium hydroxide had nanosized crystallites (239.9 Ǻ) and MgO in this sample showed much smaller crystallite size (377.7 Ǻ) than that in the thermal treated sample (3908.2 Ǻ).

For the organic-acid treated samples, different results were obtained after calcination at 800 °C in air. C-D_Formic-5_ and C-D_Formic-3_ samples had very similar phase compositions to that shown in the C-D_H2O_ sample. On the other hand, C-D_Acetic-5_ sample contained very high CaO concentration (56.80%) with a small amount of 0.25% Ca(OH)_2_.

Regarding dolomite samples treated with inorganic acids (nitric acid, sulfuric acid, or phosphoric acid), the phase composition changed significantly, and it greatly depended of the acid type and pH value. For C-D_Sulfuric-S_ sample, only pure CaSO_4_ crystals were obtained. The absence of Mg related phase in this sample probably resulted from Mg leaching. However, for C-D_Sulfuric-1_ the sample, it consisted of 53.45% Ca(OH)_2_, 11.52% CaO, 31.18% MgO and 3.85% of CaSO_4_. In case of C-D_Phosporic-S_, 67.37% Ca_2_P_2_O_7_ and 37.63% Mg_2_P_2_O_7_ phases were obtained. On C-D_Phosporic-5_ regard, its structure was similar C-D sample.

The XRD analysis and the Rietveld refinement results show that different treatments may lead to different phase composition of dolomite. Even with the same acid, but with different pH value or acid/dolomite molar ratio, the final sample composition varied significantly. The crystallite size of Ca(OH)_2_ in all the corresponding samples was in nanosize scale, ranging between 20 and 26 nm. The formation of 5~13% CaCO_3_ was related to the reaction of CaO with CO_2_ in air.

### 3.2. FT-IR Characterization of the Treated Samples Before Calcination

Figure 3a shows the FT-IR spectra of the uncalcined dolomite and the samples treated with water or acids. Calcium and magnesium hydroxides, carbonates and oxides were identified for the untreated parent dolomite sample (U-D). The intense peak at 3642 cm^−1^ is assigned to the O–H stretching vibration mode of Ca(OH)_2_. Four vibrational IR modes can be observed for the free CO_3_^2−^ ion: (i) the symmetric stretching, at around 1080–1085 cm^−1^, *ν_1_*[CO_3_]; (ii) the out-of-plane bend, at around 700–720 cm^−1^, *ν_2_*[CO_3_]; (iii) the asymmetric stretching, at around 1420–1580 cm^−1^, *v_3_*[CO_3_]; (iv) the split in-plane bending vibrations, at around 850–880 cm^−1^, *ν_4_*[CO_3_] [18]. Therefore, split absorption bands in the range of 1400–1500 cm^−1^ was attributed to the antisymmetric stretching vibration of the O–C–O groups of carbonates. The weak band near 875 cm^−1^ was assigned to the in-plane bending vibrations of C–O bond in carbonates [19].

The sample U-D_H2O_ shows sharp bands at 3690 and 3642 cm^–1^ (Figure 3a), suggesting the presence of magnesium and calcium hydroxides, respectively. These two bands resulted from Ca(OH)_2_ and Mg(OH)_2_ due to CaO or MgO reacting with water (the main reactions were shown in Table 3). The split absorption band in the range of 1400–1500 cm^–1^ and a band around 875 cm^–1^ correspond to distorted carbonates.

The FT-IR spectrum of U-D_Formic-S_ sample is very similar to U-D_H2O_ except for the small band appearing at 1050 cm^–1^. It seems to indicate the presence of the vibration of C=O bond in the calcium formic acid species or the symmetric stretching vibration of carbonate, *ν_1_*[CO_3_]. The reactions 2 and 3 (Table 3) could take place during the formic acid treatment. However, under the experimental conditions, the thermodynamic parameters ΔH°_r_ and ΔG°_r_ suggest that these reactions between calcium oxide and formic acid could not occur, thus, only formic acid molecules were adsorbed on the sample surface without intermediate compounds.

For the U-D_Acetic-S_ sample, acetic acid could react with CaO or Ca(OH)_2_ to produce calcium acetates (reactions 4 and 5 in Table 3). These reactions were confirmed by the bands at 2965, 1600, 1535, 1428, and 620 cm^–1^, which belong to calcium acetate [20]. In this sample, the hydroxyls bands almost disappeared, indicating the elimination of Ca(OH)_2_ via reaction with acetic acid.

In the FT-IR spectrum of sample U-D_Nitric-S_, the bands at 1630, 1320, 1012, and 820 cm^−1^ correspond to calcium nitrates produced via reactions 6 and 7 in Table 3. While the band at 3642 cm^–1^ indicates the presence of hydroxyl groups.

Treatments with sulfuric and phosphoric acid resulted in reactions 8–9 and 10–11 taking place, respectively (Table 3). The FT-IR spectra in Figure 3a indicate the presence of sulfates and phosphates in samples U-D_Sulfuric-S_ and U-D_Phosphoric-S_, respectively. Similar analyses were performed on the samples treated with various acid solutions at different pH values (1, 3, and 5). The main observations are summarized as follows: (i) the samples treated with solutions at pH = 1 (U-D_Sulfuric-1_, U-D_Phoshoric-1_, and U-D_Acetic-1_) exhibited related bands to sulfates, phosphates, and acetates, respectively; (ii) associated signals of Ca and Mg hydroxyls were detected for both samples, pH 3 and 5.

### 3.3. FT-IR Characterization of the Treated Samples after Calcination

Figure 3b shows all the FT-IR spectra of the different calcined samples. For the C-D sample, the FT-IR spectrum exhibits a weak band at 3642 cm^–1^, indicating most of the surface hydroxyls were removed. Even though C-D_H2O_ and C-D_Formic-S_ presented the same band, their intensity was higher, which could be caused by the rehydration of oxides related to ambient moisture. The signals identified at 1400–1500 cm^−1^ suggest the CO_2_ adsorption [22].

The FT-IR spectra of sample C-D_Acetic-S_ shows bands corresponding to –OH (3642, 1650 cm^−1^), CO_3_^2−^ (1500–1600 cm^−1^), and C–O (875 cm^−1^). However, the signal for calcium acetate showed in the U-D_Acetic-S_ sample (Figure 3b) was not present for the C-D_Acetic-S_ sample, after thermal treatment.

For the C-D_Nitric-S_ sample, the bands at 1500–1400 and 875 cm^−1^ indicate the presence of carbonate result from the recarbonation in air and nitrate phases. The sample C-D_Sulfuric-S_ showed a band at 1100 cm^−1^, corresponding to the antisymmetric stretching of metallic sulfates, and bands at 670 and 609 cm^−1^, which are assigned to the bending modes of sulfate [23].

The IR spectrum of sample C-D_Phosphoric-S_ showed signals related to calcium phosphate (536, 558, 1027, 1071, and 1138 cm^−1^), along with a band at 724 cm^−1^, corresponding to calcium pyrophosphate (Ca_2_P_2_O_7_) [24].

In the spectra of samples C-D_Nitric-S_, C-D_Sulfuric-S_, and C-D_Phosphoric-S_, the bands of hydroxyls are absent. In these samples, primarily calcium nitrate, sulfates, and phosphates were present after calcination.

### 3.4. Thermogravimetric Analysis (TGA) of Uncalcined Samples

Figure 4 shows the TGA profiles of the different uncalcined samples. The U-D sample showed two weight loss stages at 350–450 °C and 600–700 °C (Figure 4a). The former corresponded to the decomposition of Mg(OH)_2_ and Ca(OH)_2_, whereas the latter to the decomposition of MgCO_3_ and CaCO_3_ into MgO and CaO.

The U-D_H2O_ sample showed three weight loss stages. Two in the ranges of 315–395 °C and 400–480 °C that corresponded to the decomposition of Ca(OH)_2_ and Mg(OH)_2_, respectively [25], in agreement with the FT-IR results. The third stage (500–700 °C) was associated to decomposition of MgCO_3_ and CaCO_3_ into MgO and CaO. Sample weight became constant above 700 °C, suggesting that only oxides were present [26]. The TGA profile of sample U-D_Formic-S_ is almost identical to that observed on the sample U-D_H2O_. As explained in Section 3.2, formic acid treatment did not modify the phase components.

There are four weight loss stages in the TGA profile of U-D_Acetic-S_ sample, indicating complete decomposition procedure because the acetate decomposition was accompanied by the transformation of hydroxides into oxides. The first two stages corresponded to the decomposition of Mg(OH)_2_ and Ca(OH)_2_, respectively, whereas the third stage indicated the decomposition of acetates to form carbonates species between 315 and 490 °C. Moreover, the decomposition of carbonates to oxides occurred in the range of 620–770 °C.

In U-D_Nitric-S_ thermogram (Figure 4b), the decomposition of hydroxides and nitrates occurred in the range of 270–530 °C [27]. Based on the change in slope observed at ~400 °C, both compounds were decomposed directly to CaO and MgO. For the U-D_Sulfuric-S_, only water loss was detected (100–150 °C) since calcium species reacted with sulfuric acid to produce thermally stable sulfate compounds. U-D_Phosphoric-S_ sample presented two additional weight loss stages, corresponding to phosphoric acid (250 °C) and hydroxides (450 °C) decompositions. These results indicate that phosphoric acid partially reacted with calcium and magnesium species to produce phosphates, which means that hydroxides were also produced. Neither sulfate nor phosphate decomposition were observed in samples U-D_Sulfuric-S_ and U-D_Phosphoric-S_ since the maximum TGA operation temperature is 1000 °C and calcium sulfate and phosphate decompose at ~1200 °C [28] and ~1400 °C [29], respectively.

Samples U-D_Acetic-S_ and U-D_Nitric-S_ samples showed the greatest weight loss, approximately 60 wt% and 45 wt% respectively. A slight weight loss (~5 wt%) was observed for U-D_Sulfuric-S_ at ~150 °C which attributed to the desorption of water.

Three outcomes were observed after dolomite was treated with stoichiometric acid solutions: (i) For C-D_H2O_ and C-D_Formic-S_, only water reacted to form Ca and Mg hydroxides; (ii) C-D_Acetic-S_ and C-D_Nitric-S,_ acid reacted with Ca and Mg and the intermediates were thermally decomposed into CaO and MgO; (iii) On C-D_Sulfuric-S_ and C-D_Phosphoric-S,_ acid reacted with Ca and Mg and the corresponding compounds were thermally stable.

Similar analyses were performed on the samples treated with acid solutions at different pH values. Results are summarized as follows: samples treated at pH = 1 (U-D_Sulfuric-1_, U-D_Phoshoric-1_, and U-D_Acetic-1_) exhibited sulfates, phosphates, and acetates, respectively. After thermal treatment, no evidence of organic or nitric groups were found in samples C-D_Acetic-1_ and C-D_Nitric-1_ as confirmed by the FT-IR analysis. Similar to the stoichiometric treatments, sulfates and phosphates were found in samples C-D_Sulfuric-1_ and U-D_Phoshoric-1_. Moreover, U-D_Formic-1_ sample was not included in the analysis because it was not possible to obtain a formic acid solution with pH 1.

The TGA thermograms of the samples treated at pH 3 and 5 were remarkably similar. The TGA results showed evidence of the decomposition of Mg(OH)_2_ and Ca(OH)_2_ at 300–400 °C and 400–500 °C, respectively, along with the decomposition of residual carbonates into oxides, at 500–700 °C.

### 3.5. Textural Properties of the Treated Samples

The textural properties after acid and thermal treatments are reported in Table 4. A type-IV isotherm characteristic with a hysteresis loop related to the mesoporous structures was exhibited on U-D, C-D, and C-D_H2O_ (Appendix A) [3]. Among them, U-D presented the lowest adsorbed volume and surface area with a negligible hysteresis. After thermal treatment, the average pore size of C-D sample did not show relevant change; however, its pore volume and surface area increased by approximately two times. The surface area of sample C-D_H2O_ increased approximately 72% compared with C-D sample. This increase was related to both, the unclogging of pores and structural rearrangement promoted by the formation of intermediate hydroxyl groups (as discussed on Section 3.1) which provided negative charges that prevent agglomeration [30].

After the stoichiometric acid treatments, the samples C-D_Formic-S,_ C-D_Acetic-S_ and C-D_Nitric-S_ also exhibited type-IV isotherms with H3-type hysteresis loops. The surface areas of C-D_Formic-S_ and C-D_Acetic-S_ samples increased by approximately 5 times with respect to C-D. The formation oxides from hydroxide and acetate intermediates resulted in a uniform structure and pore size distribution, leading to the observed increases in surface area [31]. However, exfoliation might also have affected the surface area, especially in the case of formic acid [32]. In contrast, the surface area of sample C-D_Nitric-s_ was slightly higher in a factor of 0.37 indicating that the conversion of nitrates to oxides did not significantly increase the surface area.

The samples C-D_Sulfuric-S_ and C-D_Phosphoric-S_ showed type-II isotherms characteristic of nonporous or macroporous adsorbents. These samples exhibited the lowest adsorbed volumes, and their isotherms did not show hysteresis loops (Appendix A). Consequently, small surface areas (<7 m^2^/g) took place. This effect is explained by the stability of sulfates and phosphates, which remained stable at the calcination temperature (800 °C) [33].

For the samples treated at pH 1, the largest surface area was obtained in sample C-D_Acetic-1_ (~31 m^2^/g) while for inorganic-acid treatments, surface area slightly increased (19–22 m^2^/g). This result could be was due to the conversion of nitrates into oxides and the presence of sulfates and phosphates.

Similar behaviors were observed for pH 3 and 5 treatments. The largest surface area was obtained with formic acid at pH 3 (46 m^2^/g) whereas, all samples treated at pH 5 exhibited type-IV isotherms with H3-type hysteresis loops (Appendix A), and pore sizes were between 3.5 and 4.5 nm. The surface area of C-D_Acetic-5_ and C-D_Nitric-5_ (~45 m^2^/g) was comparable to the highest surface areas observed at low pH (C-D_Formic-S_, C-D_Acetic-S_, and C-D_Formic-3_). As the pH value was closer to 7, the isotherms were similar between them and had greater adsorbed volume than the C-D_H2O_ sample_._

Under the experimental conditions, the different treatments modified the surface area and pore volume of dolomite. The change in surface area depended on the intermediate compounds produced. Higher increases in surface area were observed for solutions at pH 5 due to the formation of hydroxides as intermediates and structural rearrangement. This resulted in a regular pore size distribution and the elimination of pore blockage. The number of negative charges was increased by the generation of hydroxyls that avoid agglomeration during thermal treatment [30].

### 3.6. CO_2_ Adsorption and Basicity

The basicity and base strength of the treated dolomite samples were measured by CO_2_-TPD method and the results are shown in Figure 5a. The CO_2_-TPD profile of C-D sample exhibited two small peaks: one at between 240 and 350 °C and another at between 470 and 540 °C. The first peak could be assigned to interaction of CO_2_ with the weak basic sites which could be related to bicarbonates generated from interaction of CO_2_ and OH^−^ groups. The second peak associated to moderate-strength basic sites, could be related to bidentate carbonates formed with the oxygen atoms of Mg^2+^–O^2−^ and Ca^2+^–O^2−^ pairs, [34]. Comparable basic sites were presented at similar temperatures ranges for the samples C-D_H2O,_ C-D_Formic-5_ and C-D_Nitric-5_, among these, C-D_H2O_ presented the higher desorption peaks.

The other treated samples presented one type of basic site. For instance, in C-D_Sulfuric-1_ and C-D_Sulfuric-S_ the weak basic sites were eliminated, and the amount of CO_2_ adsorbed on the moderate-strength sites increased. On contrary, in samples C-D_Formic-3_ and C-D_Acetic-S_ the moderate-strength sites were removed, but the weak basic sites were enhanced. Treatments with phosphoric acid eliminated almost all the weak basic sites in the solids.

Figure 5b shows the CO_2_-TPD profiles in high temperature region in range 600-800 °C where a third CO_2_ desorption peak was presented in most of the samples. This signal was previously assigned to strong basic sites, likely corresponding to CO_2_ adsorbed at isolated O^2−^ in a particular position [35]. This CO_2_ desorption peak may indicate thermal decomposition of residual and new formed carbonates according to the desorption temperature along with the TGA and FT-IR results. The C-D_Phosphoric-S_ and C-D_Sulfuric-S_ samples did not contain any strong basic sites as evidenced by the complete absence of the CO_2_ desorption peaks. This is because these two samples were composed primarily of pyrophosphates and sulphates, respectively, as they are compounds without capacity of adsorbing CO_2_. A high desorption signal of CO_2_ at this temperature range can be observed for most of the treatments, among which C-D_Acetic-S_, C-D_Nitric-S_, and C-D_Nitric-5_ achieve the highest signals. These results showed the performed treatments on dolomite enhanced the formation of strong basic sites, which make them attractive to some applications as CO_2_ capture that requires strong basic sites.

### 3.7. Catalytic Activity—Synthesis of Glycerol Carbonate

The catalytic activity of treated dolomite was evaluated by performing the synthesis of glycerol carbonate. The apparent glycerol carbonate production rate (r_gc_) of the evaluated catalyst is shown in Figure 6. The sample without treatment C-D had a r_gc_ value 1.86 mmol/min·g_cat_ at a glycerol conversion 40.7%. The highest activity was achieved on the C-D_H2O_ catalyst with a r_gc_ value of 3.42 mmol/min·g_cat_ (75.6% conversion), followed by 3.32 mmol/min·g_cat_ (73.4% conversion) on C-D_Formic-5_, and 2.88 mmol/min·g_cat_ (63.6% conversion) on C-D_Sulfuric-1_. These values are 84, 78, and 55% greater than C-D catalyst, respectively. Thus, some acid and hydrothermal treatments could significantly improve the catalytic activity of dolomite. The catalytic activity results are five, four, and three times higher than those reported in the literature under similar reaction conditions with MgO–CeO_2_ [36], CaLa [12], and Li/CaO [37] catalysts, respectively.

Even though, samples C-D_Formic-S_, C-D_Acetic-S_, and C-D_Nitric-5_ exhibited surface areas above 40 m^2^/g (see Table 4), their r_gc_ was lower than that achieved with untreated dolomite (1.84 mmol/min·g_cat_). Therefore, the surface areas of the treated dolomite samples did not directly affect the production of glycerol carbonate because these samples have quite different phase compositions.

Likewise, the sample C-D_Acetic-S_ showed the highest strong basic sites among the catalysts as evidenced by CO_2_ desorption peak at 600–800 °C (see Figure 5b). However, the r_gc_ value is approximately 0.98 mmol/min·g_cat_, even lower than that achieved on C-D sample. This suggest that catalytic activity has not relationship with the strong basic sites of the samples.

It was found that a high catalytic performance exhibited a big number of basic sites with weak and moderate strength (see Figure 5). In contrast, the samples C-D_Phosphoric-S_ and C-D_Sulfuric-S_ exhibited the lowest r_gc_ values as a result of the smallest number of basic sites in the region between 200 and 600 °C, as indicated by their crystalline phase composition (Ca_2_P_2_O_7_, Mg_2_P_2_O_7_, and CaSO_4_). The experimental results confirmed that basicity with weak and moderate strength played a key role in the synthesis of glycerol carbonate.

Figure 7 shows the relationship between r_gc_ values and calcium hydroxide concentration in the samples after thermal, hydrothermal, organic acid and inorganic acid treatments. It was found that r_gc_ values and Ca(OH)_2_ content in the catalysts varied with the same tendency, indicating that the apparent glycerol carbonate production rate is almost proportional to the concentration of crystalline Ca(OH)_2_ in the catalysts. These hydroxyl species are related with the active basic sites that enhance the catalytic activity.

Based on previous studies and the results obtained from the present work, a plausible mechanism of glycerol carbonate formation is proposed in Scheme 1. The initial step is the reaction between one of the primary hydroxyl groups of glycerol and the weak-moderate basic sites (B) on the surface of catalyst, that leads to generation of a glyceroxide (CH_2_OH-CHOH-CH_2_O^−^) as an intermediate anion conjugated acid (B···H^+^). Then, the glyceroxide anion attacks the carbonyl carbon of dimethyl carbonate to form methyl glyceryl carbonate (CH_2_OH-CHOH-CH_2_-CO_3_-CH_3_) and methoxide anion as intermediates. The methoxide anion further reacts with the conjugated-acid base to recover the active sites of catalyst and produces a methanol molecule. In the final step, methyl glyceryl undergoes cyclization, and the secondary hydroxyl group reacts with the carbonyl carbon via nucleophilic attack to produce glycerol carbonate and another methanol molecule [38].

The weak- and moderate-strength basicity of the catalysts was a dominant factor in glycerol transesterification to produce glycerol carbonate. After calcination, calcium hydroxides were formed as a result of the incomplete reconstruction and rehydration of CaO, causing an increase in moderate-strength basicity, which improved their catalytic activity [39]. The C-D_H2O_ sample presented the highest number of hydroxides in the crystalline structure of the solid, and therefore, exhibited the best catalytic activity.

The surface charges produced from structural carbonates is another important factor in the basicity dependent of the treatments. High pH may favor the formation of HCO_3_^−^ and CO_3_^2−^ species, while low pH favors the presence of Ca^2+^, CaHCO_3_^+^, and CaOH^+^ species.

Furthermore, dolomite decomposition could be highly affected by low hydroxyl groups in the structure. This could be a consequence of a hydrolysis or leaching of Ca^2+^ or Mg^2+^ out of the crystalline structure.

This suggest that the best treatment to generate hydroxyl groups and therefore improve basicity, is the hydrolysis and a moderate generation of Ca^2+^ from residual carbonates, without the generation of intermediate groups, as observed with the hydrothermal treatment and with formic acid at pH 5.

Finally, among the samples, hydrothermal treatment is a green, economic, and easy way to obtain dolomite as a basic material with high catalytic activity to produce glycerol carbonate.

## 4. Conclusions

Natural dolomite materials were modified by thermal, hydrothermal, and different organic/inorganic acids treatments at three pH values (1, 3, and 5). Their structural and physical-chemical properties were characterized and their potential use as catalyst for the production of glycerol carbonate was evaluated. Several conclusions have been drawn:Different treatments led to the reorganization of the pore system of dolomite by diminishing pore diameter from approximately 17 Å to 4~6 Å and enhancing the surface area by 4~5 times. Hydrothermal and formic acid treatments greatly improved the surface area.The use of the different treatments and calcination of the dolomite modified the crystalline structure and composition, determined by the XRD analysis and Rietveld refinements. Most of the samples contained crystalline CaO and Ca(OH)_2_ in different percentages, resulting the hydrothermal treated sample with the higher content of Ca(OH)_2_. The presence of the hydroxyl species in the structure and surface could be as a result of the fast rehydration of highly active sites of CaO with moisture.The presence of Ca(OH)_2_ in the materials was related to the presence of active sites needed to catalyze the glycerol carbonate synthesis as evidenced by the fact that glycerol carbonate production rate was proportional to the amount of Ca(OH)_2_. Among the treatments the hydrothermal led to the highest concentration of active sites with weak and moderate strength basicity. However, the sulfuric and phosphoric acids at stoichiometric molar ratio produced inactive crystalline CaSO_4_, Ca_2_P_2_O_7_, and Mg_2_P_2_O_7_ phases for the glycerol carbonate synthesis, evidenced by the Rietveld refinement and FTIR results.The best catalytic performance for the synthesis of glycerol carbonate was achieved on the C-D_H2O_ and C-D_Formic-5_ catalysts with apparent carbonate production rate of 3.42 and 3.32 mmol/min·gcat, respectively. Even though most of the materials increased their superficial area, which could be related with the catalytic activity, the obtained results suggest that the key parameter required for good catalytic activity was the formation of more basic sites with weak–moderate strength in the catalysts.The hydrothermal treatment has been proven to be an economic and environment friendly method for obtaining dolomite with potential use in processes that require a weak-medium basic catalyst for the transesterification reaction of glycerol and dimethyl carbonate. These may lead to new applications of natural dolomite minerals.

## Data Availability

Not applicable.

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
