# Peer review of "Modified Natural Dolomite and Its Influence on the Production of Glycerol Carbonate: Effects of Structural and Basicity Properties"

_materials, 2021, doi:10.3390/ma14092358_

Round 1

Reviewer 1 Report

The work is well done and the manuscript is rich in experimental data, however to some extent it is descriptive. In particular, the conclusions do not appear to be conclusive enough in respect to the title and goal of the article.
This can be improved. 

Author Response

Point 1. The work is well done and the manuscript is rich in experimental data, however to some extent it is descriptive. In particular, the conclusions do not appear to be conclusive enough in respect to the title and goal of the article. This can be improved.

Response 1: Thank you very much for your comments, we improved the conclusion section in order to have a clear connection with the title and goal of the paper. This includes the relationship of the structural and basicity characteristics of the samples after the treatments and reaction to produce glycerol carbonate.

Reviewer 2 Report

Utilization of dolomite as the catalyst for production of glycerol carbonate is a green and economic strategy. In this manuscript, screening of the pretreatment methods was carried out. The changes on compositions and textural structures of the treated dolomite were discussed. In the tested pretreatments, hydrothermal is the best dolomite treatment strategy for glycerol carbonate synthesis. However, some statement in this manuscript is not supported by solid evidence. For example, a mechanism for glycerol carbonate production was proposed in this work. Nevertheless, no intermediate species was detected. Meanwhile, the active sites are signed as Ca(OH)2, CaO and MgO in Scheme 1, however, it is an unclear statement and contrary to the conclusion that Ca(OH)2 is the active site for this reaction. Furthermore, reusability is not tested in this work which is an important aspect for the hetergeneous catalyst. Finally, the language should be rewritten concisely. Overall, this manuscript provided a green and economic strategy for production of glycerol carbonate. I believe this work can be considered for publication in Materials after major revision.

Reviewer 3 Report

The Introduction fully presents the literature review and describes the aim of researches. The applied methods of dolomite samples characterization  and the catalytic reaction conditions were described in details. Going to the Results the XRD analysis were made especially minutely showing wide range of the obtained the phase composition of dolomite.

I have only few remarks:

  1. In abstract (line 15) and in Conclusion (line 535) remove the number “21”
  2. Lines 115-116 “dolomite ……was provided by a local supplier”, if you know please write the company which the extraction of dolomite.
  3. Give the source (literature) of data (Table 3) concerning the thermodynamic parameters ΔH°r and ΔG°r suggest
  4. In Fig.3 will be better entitle OX axis as “Transmittance (a.u.)”
  5. At line 274 should be “Table 3”
  6. I propose to reorganize Fig.3 and Fig4 , they should be side by side, in parallel. The order of samples in both figures should be identical. It will be easier to compare samples U-D… and C-D…
  7. Figure 5 indicates the loss of mass upon heating (DTG) whereas in the text you discuss the decomposition of samples so it is necessary to present also DTA curves (with exothermic peaks) for all samples with exact temperature of their decomposition.
  8. 6 a) concerns temperature range 100-600 C whereas Fig.6b) 500 -1000 C, the figs. are not consistent. Both figures contain the range 500-600 C but peaks which are seen in Fig.6a) (example: peak C-DSulphuric-S at 450-550C) are not visible in Fig.6b).
  9. Lines 469-470: “The catalytic activity results are 4.95, 3.93 and 3.38 times higher than those reported in the literature under similar reaction conditions with MgO–CeO2 [35], CaLa [12], and Li/CaO [36] catalysts, respectively.” write “5, 4 and 3 times higher than…”
  10. In my opinion Figure 8: “Correlation of Ca(OH)2 crystalline volume content and apparent glycerol carbonate production rate” can be replace by the relationship between: Ca(OH)2 content and rate, which will be the linear correlation.

I have the impression that the obtained results are characteristic only for natural dolomite (locally mined). In my opinion it will be valuable to publish this paper in Materials after minor revision.

Round 2

Reviewer 2 Report

The present version can be accepted.

Author Response

Dear reviewer 2, We thank you for all the comments made to improve the quality of the manuscript.